# Mesoporous Silica Particles as Drug Delivery Systems—The State of the Art in Loading Methods and the Recent Progress in Analytical Techniques for Monitoring These Processes

**DOI:** 10.3390/pharmaceutics13070950

**Published:** 2021-06-24

**Authors:** Katarzyna Trzeciak, Agata Chotera-Ouda, Irena I. Bak-Sypien, Marek J. Potrzebowski

**Affiliations:** Centre of Molecular and Macromolecular Studies, Polish Academy of Sciences, Sienkiewicza 112, 90-363 Lodz, Poland; ktrzecik@cbmm.lodz.pl (K.T.); achotera@cbmm.lodz.pl (A.C.-O.); sypieni@cbmm.lodz.pl (I.I.B.-S.)

**Keywords:** mesoporous silica nanoparticles, drug delivery systems, drug loading methods, solid-state NMR, thermal analysis, gas sorption, vibrational spectroscopy, mass spectrometry, electron microscopy, PXRD

## Abstract

Conventional administration of drugs is limited by poor water solubility, low permeability, and mediocre targeting. Safe and effective delivery of drugs and therapeutic agents remains a challenge, especially for complex therapies, such as cancer treatment, pain management, heart failure medication, among several others. Thus, delivery systems designed to improve the pharmacokinetics of loaded molecules, and allowing controlled release and target specific delivery, have received considerable attention in recent years. The last two decades have seen a growing interest among scientists and the pharmaceutical industry in mesoporous silica nanoparticles (MSNs) as drug delivery systems (DDS). This interest is due to the unique physicochemical properties, including high loading capacity, excellent biocompatibility, and easy functionalization. In this review, we discuss the current state of the art related to the preparation of drug-loaded MSNs and their analysis, focusing on the newest advancements, and highlighting the advantages and disadvantages of different methods. Finally, we provide a concise outlook for the remaining challenges in the field.

## 1. Introduction

Drug nanoformulations have been a challenge for scientists for many years. It remains very difficult to create non-toxic, body friendly systems that would deliver the desired drug concentration to the target site overcoming all barriers in vivo. One of the attractive options that can help solve this problem is the use of smart, intelligent drug delivery systems (DDS). Mesoporous silica is a promising material in this respect. Discovered in the early 1990s, mesoporous silica nanoparticles (MSNs) have found numerous applications in industry and many fields of science, including heterogeneous catalysis, electrochemistry, analytical chemistry, molecular biology, and others. The first application of these nano-structured materials in pharmacy dates back to 2001 when Vallet-Regi described the use of MCM-41 silica in the ibuprofen release process [1]. This seminal work initiated the dynamic development of research focused on the application of mesoporous particles as carriers in drug delivery systems. Silica has been approved by the FDA (The US Food and Drug Administration) as “Generally Recognized as Safe” (GRAS) and can also be used in cosmetics and as a food additive [2]. Studies in a simulated body fluid have shown that mesoporous silica nanoparticles undergo a three-stage degradation after administration, which seems to be beneficial for the drug release process. Moreover, in vivo absorption, distribution, and excretion studies in mice following administration of MSN by various routes indicate good tissue biocompatibility for oral and intravenous injection. In contrast, MSNs were difficult to absorb after intramuscular and subcutaneous administration. In vivo mortality, clinical, pathological, and blood chemistry studies showed low toxicity of MSN in mice. The silica nanoparticles accumulated mainly in the liver cells, from where they were removed within about four weeks, although liver damage has been seen at higher doses and longer use [3,4,5,6].

The treatment of a patient with synthetic drugs or natural products is a multi-stage procedure in which we can distinguish several key steps, starting from the synthesis or isolation of a chemical compound, its full characterization, approval by the FDA and/or EMA (The European Medicines Agency), and ending with the full recovery of the patient. The complexity of this process in graphical form is shown in Scheme 1. Our review focuses on the issues described in Panel 2, “Drug formulation”, and summarizes the latest developments in research into mesoporous silica particles used as drug transporters. Particularly, we will describe loading procedures and physicochemical methods utilized to understand the molecular state of the drug in mesoporous silica and its location in the pores.

## 2. Drug Loading Methods

Today the porous silica matrices are commonly used as carriers for active substances. For pharmaceutical applications, the most prevalent is the amorphous silica, due to its lower toxicity in comparison to the crystalline form. Therefore, various types of it have been investigated thoroughly [7,8,9,10,11,12], including porous and non-porous silica, fumed silica, silica gels, etc. The properties of porous silica materials can be improved further by the tailored control of pore structural features, resulting in ordered mesoporous particles, currently the most commonly used type, as demonstrated by the extensive literature display [10,13]. Due to the orderly porous structure (mostly channels without interconnection), large surface area (more than 700 m^2^/g), and high pore volume (above 1 cm^3^/g), they offer precise load control and drug release kinetics [14].

MSNs have two functional surfaces: an inner cylindrical formed by pores and an outer surface. Both surfaces can be selectively functionalized to improve loading and release properties. The outer surface can also be functionalized to increase the efficiency of delivering a drug to a specific target in the body [15,16]. The greatest advantage from the pharmaceutical point of view is the improvement of the solubility of the API (active pharmaceutical ingredient) loaded into the mesoporous silica and the formation of the drug-carrier dispersion systems [17]. Scheme 2 shows the most commonly used MSNs in the formulation of drug substances.

The loading of drugs into the pores of mesoporous silica can be accomplished in many different ways (Figure 1). Generally, they have been divided into two main approaches: solvent-free methods and solvent-based methods (so-called wet methods) [17,18].

According to the literature search, the solvent-based methods such as solvent evaporation, adsorption, and incipient wetness impregnation, which encapsulate various types of drugs are commonly employed. Among the solvent-free methods, melting and co-milling are particularly preferred to load the drug into MSNs pores (Table 1). The drug encapsulation process involves adsorption of the drug molecules to the mesoporous silica inner/outer surface under conditions that favor drug-carrier interactions like van der Waals interactions, hydrogen bonding, electrostatic binding, or covalent bonding [19]. The ideal drug loading process should quickly load a large amount of the drug and then unload it with the desired release profile [18]. It is also desirable to minimize the fraction of the drug adsorbed on the external particle surface to avoid the possibility of its crystallization due to the lack of (pores) entrapment effects [20]. The extent of drug loading depends on several parameters. The most significant is the surface area and the affinity of the drug to the silica substrate [21,22]. In the wet methods, the internal pore volume of the silica, polarity of the solvent, and the concentration of the drug in the solution are also important [23]. The choice of drug loading method affects the degree of drug loading, the distribution of the drug in silica, and the physicochemical properties of the loaded drug [9].

Successful commercialization of any DDS requires an easy, cost-effective, and reproducible method of large-scale production. Since API encapsulation is the most critical step in mesoporous silica based delivery system applications, there is a great need to develop suitable large-scale drug loading techniques. Many of the methods listed in Table 1 do not offer optimal and universal solutions that would make them indisputably used on a larger scale than in the laboratory. These restrictions significantly affect the strategy and decisions of the pharmaceutical industry, which expects simple, cheap, and repeatable solutions that meet economic and ecological criteria. The most common methods of drug loading will be characterized below, taking into account their advantages and disadvantages.

### 2.1. Solvent-Based Methods

There are several solvent-based approaches for loading a drug into mesoporous silica. All of them improve drug release through the amorphous state of API in MSN. The most popular are described below. In the wet methods, regardless of the used procedure, it is necessary to remove the solvent to the acceptable levels specified in the guidelines of the International Conference on Harmonization (ICH) Q3 (R5) [75]. It is worth mentioning that some residual solvents used may remain in MSN despite its disposal. As demonstrated by Eren et al. the hexane used in the synthesis of SBA-15, although later evaporated, reduced the viability of the HLA-116 and HT-29 cells incubated with these particles [34]. Therefore, using less toxic and safe solvents, e.g., ethanol or supercritical or near-critical CO_2_, as an alternative to organic solvents in the drug loading process is recommended, if possible, for human pharmaceutical applications [34,52,76]. The loading procedures using solvent methods are challenging because it often requires multi-stage and long-lasting processes, uses large amounts of solvents, and it is also difficult to control the filling factor. Another, equally important issue is the concentration of the drug used and the selection of an appropriate solvent.

*The adsorption method* is a simple and widely used technique for loading drugs into the pores of MSNs. In adsorption from the organic solution, the mesoporous silica is immersed in the concentrated drug solution. After this time, the drug molecules are adsorbed on the pore walls. The drug-loaded MSNs are then separated from the solution by filtration or centrifugation. Finally, the particles are dried to remove the residual solvent. However, sometimes this step is omitted [77]. The adsorption method can be used for both hydrophilic [78] and hydrophobic [79] drugs. The process does not require high temperatures, so it is suitable for loading thermally sensitive substances. However, in most cases this process is inefficient, therefore a high concentration of drug solutions is required to achieve a high degree of drug loading. This can be problematic, especially for drugs with poor solubility [77]. If the concentration is too high, the drug molecules will quickly adsorb onto the surface and block the mesopores, reducing the potentially available surface area for drug loading [80]. The main disadvantage of the adsorption method is time-consuming. It is also difficult to predict what degree of drug loading will be achieved. Moreover, a large proportion of the drug is wasted in the filtration/centrifugation process [81]. The choice of a solvent has a significant influence on the process of loading drugs into MSN. The solvent with the highest API solubility is not necessarily the best candidate for obtaining a high filling factor. For example, valsartan dissolved in dichloromethane can be incorporated into MSN with a twice higher degree of loading than methanol, although the concentration of valsartan in methanol was higher than in dichloromethane. Similarly, the aprepitant loaded into MSN using chloroform as a solvent obtained over three times higher filling factor than at using methanol, even though its concentration in chloroform was approximately three times higher compared to methanol [82]. Charnay et al. showed that the use of polar solvents such as dimethyl sulfoxide, dimethylformamide, and dimethylacetamide causes a low degree of ibuprofen loading into MCM-41 [54]. In contrast, ethanol and hexane (non-polar solvents) gave relatively high filing factors. The hexane loading capacity was up to 37 wt %. This is because polar solvents can form competitive adsorption with drug molecules, causing a low degree of drug loading [54]. If the affinity of solvent (often associated with the formation of hydrogen bonds) to MSN is higher compared to the drug, the pores of the silica are filled by solvents and there is not enough space for API [83]. This is one of the limitations of the applicability of this method.

*The incipient wetness impregnation method* is the most commonly used for the preparation of catalysts [84,85] but is also suitable for loading drugs into mesoporous materials [29,54,57]. In this method, a known volume of concentrated drug solution approximately equal to the pore volume of the MSN used is added dropwise to the silica. Next, the wet powder in which the drug diffuses into the pores by capillarity is dried and quickly washed on a filter with a small amount of used solvent to remove the excess drug coating the external surface of silica [54]. Finally, the powder is dried using air for 24 h and placed under a vacuum for 48 h at a temperature of 40 °C [57]. The washing step for eliminating APIs outside the pores has a significant impact on drug loading efficiency. Charnay et al. observed reducing the amount of ibuprofen encapsulated onto MCM-41 after washing the silica with ethanol from 1350 to 500 mg/g [54]. They also proved that successive impregnations enable to achieve a complete pore filling and significantly improve the concentration of ibuprofen. Their research has shown that the incipient wetness impregnation method is more efficient than the traditional solvent immersion process. Ibuprofen in the pores was in an amorphic state and the dissolution rates were increased compared to the crystalline drug [54]. The main advantage of the impregnation method is that it exploits the large pore volume of the mesoporous silica and can easily determine the amount of drug-loaded into the carrier [57]. Furthermore, the drug is efficiently loaded and thus the method is suitable for expensive drug molecules. However, to obtain a high degree of drug loading, typically in the range of 5–40% *w/w*, the high concentrated drug solution and repeated impregnations are required, which significantly extend the process of loading the drug into the MSN [81]. The uniformity of drug distribution is difficult to control. There is also a high risk of drug crystallization on the outer silica surface and plugging the mesopores [20].

*The solvent evaporation method* is another often used drug loading approach that involves a combination of adsorption and subsequent rapid solvent evaporation [86]. In this method, the silica is dispersed in a drug volatile organic solution (e.g., ethanol, dichloromethane) and dried by fast solvent evaporation using a rotary evaporator [65] or heating [49] for obtaining drug-loaded MSNs [20,87]. In the adsorption method, the solvent is removed by filtration, which can result in partial loss of the drug. In contrast, solvent evaporation may affect the physical state of the drug, its localization in the MSN, and the rate of release from the carrier [57,86]. Mellaerts et al. used the solvent evaporation method and incipient wetness impregnation to load itraconazole and ibuprofen into SBA-15 silica [57]. In both methods, the samples were dried at 35 °C and then placed under reduced pressure (1 hPa) at 40 °C for 48 h. The ibuprofen location in SBA-15 was the same regardless of the technique used. Ibuprofen preferably was positioned inside the micropores. It was observed that itraconazole molecules were located on the walls of the mesopores during the adsorption process from dichloromethane, while the incipient wetness impregnation method favored the positioning of itraconazole in the micropores (Figure 2). The authors suggested that the localization of itraconazole in the mesopores was related to the slow introduction of the drug requiring diffusion of itraconazole molecules from an external dichloromethane solution into the SBA-15 particles. Thus, the solvent evaporation method gives the drug molecules enough time to rearrange and aggregate inside the mesopores. In the incipient wetness impregnation method, all the itraconazole molecules were incorporated at once into the pores, followed immediately by solvent removal. Samples prepared via the solvent evaporation method showed the fastest release kinetics [57].

*Diffusion supported loading* (DiSupLo) is the new extremely easy and efficient method of loading APIs into the pores of mesoporous silica developed by Potrzebowski and co-workers [47]. In this approach the starting material, pre-homogenized physical mixture of API and MSN with the desired proportion of both or more components is transferred to an opened weighing vessel as a solid matter and then placed in a closed vessel containing ethanol for 3 h at room temperature (Figure 3).

The drug loading time was established experimentally using NMR measurements at 30 min. intervals. The layer thickness of the physical mixture should be small and experimentally optimized. Finally, the ethanol was thermally removed from the drug-loaded samples. The mechanism of drug encapsulation in the DiSupLo method is as follows. In the first stage of the process, there is no direct contact between the solid sample and the liquid solvent, the only one between the diffusing ethanol vapors and the solid mixture. In the second step, the vapors penetrate the solid-state mixture in the whole volume, locally condense and dissolve the drug. At this stage API dissolved in the minimum volume of solvent is transported into the pores of MSN [47]. The applicability of the DiSupLo method has been proven by testing the loading of selected non-steroidal anti-inflammatory drugs such as ibuprofen, flurbiprofen, ketoprofen, and their mixtures to the pores of MCM-41 as the drug carrier. Studies have shown the introduction of all drugs with weight ratios API/MCM-41 of 1:3 into the pores of the silica with a filling factor of 33% and 50%, respectively. For a 1:1 ratio, it has been observed that some API was inside the pores with nearly 60% filling factor while part remained on the outer wall of MCM-41. It has been proven that the DiSupLo method, apart from one API, also allows the loading of two and three drugs simultaneously with a weight ratio of API to MCM-41 of 1:3. DiSupLo uses and enhances the advantages of wet methods and eliminates their drawbacks. It is a fast process requiring a minimum amount of solvent, so it is environmentally friendly and economically justified. It is the easiest method for loading API into MSN pores described so far in the literature, since it does not require any special equipment and/or demanding experimental conditions (high temperature, high pressure, stirring, grinding, etc.) for carrying out the process. The DiSupLo method allows to achieve a high drug loading degree and encapsulate the whole API if the weight ratio of the starting API/MSN mixture is not higher than the maximal filling factor. The unique advantage of this technique is the ability to quantitatively load two or more components with the desired API composition, which significantly extends the applicability of MSN as carriers for multi-component systems [47]. Most studies using MSN as drug delivery systems focus on loading only one drug into the pores. Attempts to introduce more complex, binary, or ternary systems are rare [88,89]. Controlled-release drug synergism can be a useful therapeutic tool, especially in cancer therapies.

*Supercritical fluid technology (SCF)* commercially used in the food and chromatography industries is also a method for drug loading into mesoporous silicas [41,45,90]. The use of supercritical fluids in this technique has many advantages compared to traditional organic solvents. Their specific properties, including liquid-like density, gas-like viscosity, a diffusivity higher than that of liquids, and a very low interfacial tension allow them to be employed as impregnating agents [91]. Then, the transport of the solubilized substance in the supercritical phase is facilitated to the solid matrix, which is impregnated evenly and with a shorter processing time than in the liquid phase [45]. Carbon dioxide (CO_2_) is the most commonly used in SCF technology as it has mild critical conditions (31.2 °C, 7.4 MPa). Besides, supercritical CO_2_ (SCCO_2_) is inert, non-flammable, non-toxic, and inexpensive, making it an excellent solvent [91,92]. Another advantage of the SCF technology is that the final product after fluid evacuation is free of the residual solvent when the drug loading process is performed without the addition of a cosolvent. Li-hong et al., used an adsorption method and SCCO_2_ technology to load ibuprofen into the pores of MCM-41 [93]. It was shown that the amount and depth of ibuprofen penetrated the pores of mesoporous silica by the SCF method were larger than those using adsorption from ethanol. The authors also studied the ibuprofen release profile and found that the sustained-release effect of the drug in the samples made with the SCF technique was 50% in 15 min and 90% in 60 min, which was longer than the one prepared by the adsorption method [93]. A similar study was carried out with fenofibrate using incipient wetness and SCCO_2_ methods [45]. Both impregnation processes were compared in terms of yield, duration, and physical characteristics of the drug. Studies have shown that the incipient wetness method to load fenofibrate up to 300 mg drug/g silica in 48 h, while the SCCO_2_ method resulted in loading up to 485 mg drug/g silica in 120 min, with a low degree of crystallinity (about 1%) comparable to crystallinity observed for the classical solvent method. SCF technology, therefore, provides an alternative and environmentally friendly way to load a drug onto silica support with the advantage of reduced processing time that can be achieved in 2 h [41,45,93]. Stam et al. demonstrated that the very low concentration of ibuprofen in a nonpolar liquid CO_2_ was enough to achieve maximum loading of this drug in the pores of MCM-41 [94]. These studies have shown that liquid CO_2_, cheaper than SCCO_2_, can be an effective “green solvent” to incorporate drugs into mesoporous silica. Ahern et al. compared the use of supercritical and liquid CO_2_ technology with other drug loading methods such as physical mixing, melting, and solvent evaporation [43]. Fenofibrate and SBA-15 silica were used in the research. All methods, except physical mixing, loaded the drug into the mesopores in a non-crystalline form. The release of fenofibrate from all samples was faster than that of the crystalline drug. Interestingly, the drug release from the sample prepared by the physical mixing method was improved but had a slower rate and a lower drug release rate than all other methods. Furthermore, the release of fenofibrate from the melted sample was slower compared to the solvent methods due to the negative effect of the drug’s molten viscosity on mesopore blocking [43]. Despite the many advantages of using SCF technology to impregnate BCS II molecules within the pores of MSN, studies showed that a lot of drugs exhibit poor solubility in SCCO_2_ [95,96,97]. Therefore, lest this issue obscure the potential of this method, it has been suggested that other properties of compressed carbon dioxide may be beneficial, such as its ability to lower the melting point of the compound and reduce the viscosity of the molten products [98].

*The one-pot drug loading and synthesis procedure* is a new approach to drug encapsulation in which API is loaded into the mesoporous carrier during its synthesis [46]. The main route of MSN production is gel-sol synthesis using inert surfactant micelles as a template, which is finally removed through calcination or extraction [19]. The obtained MSNs are loaded with a targeted drug by one of the available methods [18,20,46]. This traditional method is multi-step, time-consuming, and results in low drug uptake and its burst release, which is inadequate for most applications [99]. The new method combining MSNs synthesis and drug loading process overcomes these limitations and seems very promising for the design of novel drug delivery systems. Dementeva et al. presented the possibility of using the micelles of the bactericidal drug myramistin as a template in the sol-gel synthesis of mesoporous silica nanocontainers [100]. The synthesized nanocontainers were characterized by a very high drug content (≥1 g/g of SiO_2_) and were also pH-sensitive. It should be noted that this method can be extended to other amphiphilic drugs (e.g., anti-cancer, anti-inflammatory, or bactericidal) as templating agents [46,101,102,103]. Interesting research was presented by Wan et al. using the one-pot synthesis method by evaporation-induced self-assembly, they introduced two drugs into mesoporous silica: hydrophilic heparin and hydrophobic ibuprofen [104]. It was shown that in situ drug loading realized through evaporation-induced self-assembly (EISA) had several advantages. The first merit was the time reduction of the delivery system preparation from about 74 h to 10 h. The second merit of in situ loading was the dramatically increased amount of heparin and ibuprofen in SBA-15 compared to the traditional solvent impregnation method. Sustained drug release over 30 days has also been achieved. Finally, the one-pot technique was environmentally friendly because the whole procedure was performed at ambient temperatures, with no calcination process, and without using toxic organic solvents, which is extremely desirable from a pharmaceutical point of view. Tourne-Peteilh et al. showed the drawback of this method may still be time-consuming and low drug loading [105]. They presented the use of sol-gel one-pot synthesis for ibuprofen encapsulation. Tween 80, a widely used surfactant in food and pharmaceutical formulations, has been chosen as a template for silica and also as a solubilizing agent for ibuprofen. In vitro studies in simulated physiological media showed a significant increase in the dissolution rate of ibuprofen compared to pure drug crystals. However, they achieved a low drug loading (<4%). Although the procedure also involved the synthesis of silica particles, it took several days to complete. Undoubtedly, in the one-pot method, selecting the appropriate surfactant will be the key to fine-tuning the drug loading, release, and interaction with the silica walls.

*Covalent grafting* is another possibility for loading drug molecules into the mesoporous silica [89,106,107]. Covalent linking of the drug to functional groups present on the pore walls of a mesoporous silica carrier is an attractive alternative to physical adsorption as it prevents undesirable leaching of the drug before it reaches its target site, drug release is more controlled and can occur after breakage of the covalent bond. Commonly employed bonds for covalent attachment of drugs are amide, disulfide, ester, thiol, and carbamate bonds. The targeting and release of the drug from such systems can be achieved by hydrolysis, enzymatic degradation, exchange reaction, or redox reaction [108]. This release mechanism is often called linker cleavage and has been summarized in several excellent reviews [109,110]. The covalent linking of ibuprofen to the surface of MCM-41 functionalized with amine groups has been demonstrated [111]. The stronger amide bond between the drug and the mesoporous silica surface led to an increase in ibuprofen loading, slower and more controlled drug release compared to the adsorption method. In the case of ibuprofen adsorbed into non-modified MCM-41, the weak interaction derived from the carboxylic acid groups of the drug and the silanol groups of the silica resulted in ibuprofen was easily desorbed [111]. Other exemplary studies have reported the use of the covalent linking to load cysteine, a sulfasalazine prodrug, and paclitaxel [69,112,113]. Rosenholm et al. presented the covalent attachment of the chemotherapeutic agent methotrexate (MTX) to poly (ethyleneimine)-functionalized MSNs [114]. The covalent linking of MTX to silica by using standard EDC coupling linking the carboxylic acid group present in the drug to the amino groups of silica prevented MTX premature release, and its intracellular detachment from the carrier in vitro was due to enzymatic degradation of the amide bond in lysosomes [114]. Loading of MTX to the mesoporous carrier was preferred over physical adsorption. Previously, it was shown that MTX does not adsorb to silica, only after alumina was introduced into the silica framework the adsorption was observed [115]. Moreover, the attachment of MTX mainly to the particle surface enables the loading of additional drugs into the pore network for drug combination therapies [114]. Despite the advantages of covalent bonding over physical encapsulation, such as slow drug release, improved biodistribution, therapeutic efficacy, and reduced systemic toxicity [107], this approach has some drawbacks. The chemical bonding of the drug to the surface of MSNs can convert API to an inactive form, so it is important to perform a test confirming the original molecular structure of the drug is preserved. The presence of sufficient functional groups on the carrier surface is not always possible as they influence the colloidal stability of the carrier and its surface charge. Moreover, the steric hindrance of the drug molecules crowded on the silica surface may negatively influence the formation and breaking of chemical bonds [69,77,106].

### 2.2. Solvent-Free Methods

Compared to wet methods, solvent-free methods are less time-consuming and can achieve a high degree of drug loading. Moreover, the drug concentration in the mesoporous material used is easy to predict as it is directly influenced by the ratio between API and MSN. Undoubtedly, they are environmentally friendly techniques, as they do not require checking the residual solvent in drug products, and they are located in the stream of “zero waste” chemistry.

*The melt method* involves heating a physical mixture of the drug and mesoporous silica above the melting point of API [9,116]. This approach is very efficient and significantly reduces the time of drug incorporation into the mesoporous silica. Potrzebowski et al. showed that the melt method (which they called Thermal Solvent Free) is a much more efficient technique of confining ibuprofen in the pores of MCM-41 compared to incipient wetness [83]. From the analysis of the proton NMR spectra recorded at the very fast rotation of the sample, they have established the percentage of mesopores filled. The filing factor defined as a weight to weight ratio of ibuprofen to MCM-41 was approximately 60% [83]. Despite the advantages, the melting method has some limitations. It can be suitable only for thermally stable drugs characterized by a low viscosity after melting. Some studies emphasize the high relationship between the molten viscosity of the drug and the mesopore penetration [57,117]. Mellaerts et al. showed that itraconazole was not successfully loaded into the pores of SBA-15 due to its high viscosity in a molten state which prevented homogeneous drug distribution in the mesopores [57]. Glassy itraconazole particles accumulated on the surface of SBA-15, whereas ibuprofen has a lower molten viscosity and was loaded into the pores of SBA-15 in a non-crystalline state, achieving a rapid in vitro release [57].

*Microwave irradiation* is another method to load drugs into silica materials. In this technique, the sample is heated using a feedback system that controls the temperature during the loading process of the API into the silica nanoparticles and protects the drug from degradation [118]. Waters et al., used this method to load fenofibrate into a variety of silicas including SBA-15 and compared them to samples loaded by more traditional heating methods [119]. Moreover, all experiments were performed both in the presence and absence of water, used as the fluidization medium to aid drug-carrier interaction. It was investigated that samples prepared using microwave radiation without the presence of water as a solvent showed an increased rate and extent of fenofibrate release compared to pure drug and samples prepared by other techniques. This can be explained by the fact that the crystalline fenofibrate has changed to an amorphous state and the formulation provides enhanced drug release. Furthermore, the optimal drug: carrier ratio for this method was found to be 1:3 [119].

*Co-milling* is a commonly used technique to produce sub micrometric particles [120] as well as for solid-state amorphization [121]. Recent studies have shown that milling using a planetary ball mill is an effective solvent-free method for introducing organic compounds such as benzoic acid and 4-fluoro benzoic acid into the pores of MCM-41 [122]. This study also appeared that the guest-loading process is fully controlled and dependent on the ratio of the loaded substance and silica. The maximum capacity of loading was over 50% at a 1:1 ratio. Moreover, it has been proven that using this technique, guest molecules are located in different zones of MCM-41 particles (Figure 4) [122]. Hedous et al. demonstrated that the co-milling of SBA-15 with ibuprofen made it possible to load the drug into the porous matrix, enhancing thereby the filling degree to almost 40 wt% [59]. This shows that milling can be considered as an alternative to solvent-based drug loading methods. The co-milling technique is simple, does not require time-consuming steps, and can be applied on an industrial scale. This is extremely important in practical applications. The problem to consider during ball milling processes is the resistance of the nanocarriers to mechanical stress. This issue was well recognized for MCM-41. Abu-Zied et al. investigated the effect of ball milling on the structure, texture, and morphological properties of mesoporous MCM-41 material [123]. Their results showed that by increasing the milling time, the amorphization process of mesoporous MCM-41 material accelerates. MCM-41 retains its initial crystallinity until 30 min of milling time. A serious loss of MCM crystallinity occurs after 1–2 h of milling, while the complete loss of crystallinity of this material is noticeable after four hours of ball milling [123].

## 3. Analytical Techniques

With the growing interest in the use of MSNs materials as drug delivery agents, there is an obvious need to develop adequate analytical tools to study their size and morphology, as well as loading and release processes. Knowledge about the physicochemical properties of nanoparticles is a critical step in the construction of novel delivery systems and understanding the mechanism of host-guest interactions. Many convenient and accurate tools from a wide palette of conventional analytical methods have already been used in the structural and spectroscopic studies of API/MSN systems. (Figure 5). The most common ones are thermo-analytical methods (DSC and TGA), various types of spectroscopy (NMR, IR, Raman), gas sorption, microscopy (TEM, SEM, and AFM), and powder X-ray Diffraction (PXRD). Other methods, such as chromatography, mass spectrometry (MS) have also been applied. Since different techniques analyze different properties, none of them provides full characterization but rather only a piece of partial information, therefore only their combination represents the portrayal of the complete system.

### 3.1. Thermal Analysis

Thermal analysis is a group of calorimetry techniques in which the properties of a sample are monitored against time or temperature while the temperature of the sample within a specific atmosphere is programmed [125]. Combined data from thermo-analytical methods such as differential scanning calorimetry (DSC) and thermogravimetry (TGA) allow quantification and characterization of the physicochemical nature of the drug introduced into the pores of silica. Differential scanning calorimetry (DSC) is used to characterize the physical state and location of API in a material by observing of its melting point. It has been noticed that when the drug is confined in the pores of mesoporous silica, its melting point data and glass transition state is lower than that of the crystalline state, and the melting is observed over a wider range. While the concentration of the drug in the MSN increases, the peak of the endothermic heat of fusion decreases. When all the drug is loaded into the porous silica structure, the melting point peak of the drug does not exist, meaning that the MSN prevents the API from recrystallizing and the incorporated substance is in an amorphous form. On the other hand, the drug on the outer surfaces of the pores is considered to be a bulk material and does not show a glass transition. Therefore, DSC analysis distinguish the compound on the surface of mesoporous silica from the one contained therein (Figure 6).

With the help of DSC measurements, it is also possible to determine the pore size distribution. The method is based on the observation of the solid-liquid phase transformation inside the pores, i.e., the melting/freezing point and the melting/freezing enthalpy. The melting point/freezing point depends on the pore size and the transformation enthalpy depends on the pore volume. The method also provides information about the interaction that takes place between the porous material and the liquid [126]. Knopp et al. used DSC to quantify monomolecular loading capacity by solvent-free melting during the heat-cool-heat cycle of ibuprofen with good glass-forming ability. The overloaded drug-silica systems were analyzed at various weight ratios. When quantifying excess API, the ibuprofen that is heated for the first time fuses into the pores of the mesoporous silica. After cooling, the unadsorbed part of the drug was amorphized. In the case of easily glass-forming substances, no crystallization occurs after cooling. The particles are adsorbed and immobilized on the silica surface and do not cause another glass transition in the DSC. As a result, their quantity can be determined after the silica is fully loaded by examining the change in heat capacity [127,128]. The same authors, using the thermo-analytical method, have demonstrated experimentally and theoretically the existence of a relationship between the pore volume and the surface of mesoporous silica, which will allow the determination of the maximum carrying capacity of the drug. By using this methodological combination, optimal textural properties of the silicone carriers can be designed for each drug [129]. In five types of mesoporous silica with the same chemical composition but the different surface area, diameter, and pore volume, the loading capacity of three model drugs: paracetamol, celecoxib, and cinnarizine was determined. Based on the heat-cool-heat DSC method load capacity was compared with the theoretical load capacity calculated from the surface area and amorphous density of drugs as well as the surface area and pore volume of silica. It was concluded that monomolecular loading ability increases with increasing surface area and decreasing pore volume.

Since the inorganic carrier is more thermally stable than the organic guest molecule, weight loss from the drug degradation followed by desorption of volatile components by gradually increasing the sample temperature using TGA can be observed. Weight loss is proportional to the total drug content. Figure 7 shows a comparative analysis of two methods of ketoprofen loading into MCM-41. MCM-41: the ketoprofen composite had two distinct weight loss regions; in the region below 100 °C (approx. 5%), typical for the water evaporation, and the range from 275 to 600 °C, typical for the degradation of ketoprofen. As a result of measuring TG, the total ketoprofen content in MCM-41 was 26.9% and 32.8% for the 3:1 combinations obtained by DiSupLo and milling, respectively. To determine the ketoprofen content in the remaining cases, one should take into account the population mass loss of two ketoprofen populations: (a) bulk-like molecules and (b) ketoprofen molecules interacting with the pore walls. For the pure drug, the sharp decline in the TGA curve started at 175 °C and ended at 275 °C, indicating a one-step degradation of ketoprofen over a range of 252 °C, resulting in 99.98% weight loss. The amount of solvent remaining in the pores after loading can also be determined by the same principle. However, the thermogravimetric analysis may give erroneous results in determining whether all loaded material has left the carrier when thermostable compounds or the drug itself form strong connections to MSN [130].

### 3.2. Gas Sorption

Physical gas sorption is a well-established technique of the first choice for characterizing the textural properties of solid surfaces and their changes after various types of actions. The technique precisely determines the amount of gas adsorbed on the material, which is a direct measure of the porous properties and structure. The isotherm obtained from the adsorption measurements provides information on the surface area (SA), pore volume (PV), and pore size distribution (PSD). The measured isotherms are assessed according to the generally accepted and currently used IUPAC isothermal classification and conventional models [131,132,133]. Data on gas sorption have been collected, analyzed, and presented in numerous reports, therefore the results of the experiments can be easily compared with those previously published. In the case of mesoporous silica, changes in the native porous structure and drug-loaded absorbents can be estimated based on the course of nitrogen adsorption-desorption isotherms of type IV at 77K and a wide range of relative pressures (p/p_0_) (Figure 8).

Pore size controls of MSN can be achieved on both the adsorption branch and the desorption branch of the isotherm. For mesoporous materials, capillary condensation follows through adsorption, next by a metastable fluid state. Capillary evaporation takes place by desorption and forming a hemispherical meniscus, separating vapor and condensed capillary phase. This creates hysteresis as pores of definite size are filled at a higher pressure and evacuated at lower pressures. If the test material is fully mesoporous and encloses only non-intersecting mesoporous of a cylindrical geometry and similar size, the N_2_ isotherm type IV occurs with a hysteresis loop of type H1 according to the IUPAC classification [132,133]. Thermodynamically, the desorption branch is frequently preferred in these cases for isotherm-based mesoporous size distribution. In more realistic cases where pores are randomly distributed and interconnected, the hysteresis loop will be of the H2 or H3 type and pore size distribution from the desorption branch is often more subjective by the pore network effects than the adsorption branch (Figure 8). Among the conventional models, the Brunauer–Emmett–Teller (BET) theory is used to assess the surface area of mesoporous silica; however, it is recommended to support this procedure by the repeatability of measurement when the sample also contains micropores [19]. On the other hand, the analysis of the distribution of mesoporous sizes can be performed using the Barrett–Joyner–Halenda (BJH) models based on the Kelvin equation, although recent reports indicate low reliability of this method. Computational techniques based on the density functional theory (DFT) are recommended for further validation of experimental results [133,134].

The mesoporous silica with the defined diameter (MCM-41, SBA-15 and MCM-48, and others) act as selective sieves. Additionally, they can be easily functionalized, which influences the possibility of designing the models with the tailoring ability to penetrate the pore channels by drugs and control the release mechanism. An important point is the complementarity of pore and APIs sizes. For example, due to its small size, ibuprofen can be loaded into any mesoporous system. For other larger sized drugs, such as vancomycin or glycopeptide, or others of similar size, it is better to use SBA-15 as the first option to make sure these drugs are able to end up inside the pores. When the drug to be loaded has a size much smaller than the pore size silica, most drug molecules are not adsorbed on the pore surface because only a few of them can directly interact with the matrix surface. This essential fact investigated by Vallet-Regi et al. in an alendronate model with MCM-41 and SBA-15 matrices [135] indicates that MCM-41 can load more alendronate than SBA-15 (area = 719 m^2^/g) because in MCM-41 (area = 1157 m^2^/g) the alendronate contact area is higher than SBA-15. Molecules are trapped in the pores through interactions with the inner walls of the mesopores. Additionally, considering the reverse process of drug release will depend on the surface area of the material. Various experiments have shown that when the silica surface is very large, there is high molecular retention and as a result, a slower release of the drug compared to materials with smaller surfaces. The pore volume is also an important parameter for large particle sizes. The larger the space available for loading drug molecules, the greater the load will be [136], which also affects the release of the drug. If the pore volume is low, the matrix channel may become clogged with drug particles. Concluding, all properties of native, functionalized, or doped mesoporous silicas due to their textural properties, i.e., large surface area and internal pore volume, favor the adsorption of drugs, even of large sizes. Exemplary structure parameters of selected APIs loaded into MCM-41 and SBA-15 are presented in Table 2.

### 3.3. Microscopy

Electron microscopy offers insight into the nanoparticle’s key parameters such as morphology, external diameter, surface area, and pore architecture. Because of its ability to provide detailed structural characterization in the nanometer scale, it has emerged as a powerful tool for the investigation of various nanostructured materials, including MSNs. Despite the undeniable advantages in imaging the structure and morphology of the API/MSNs systems, microscopic techniques have also several limitations. Its main disadvantage is the destructiveness and presence of undesirable artifacts from sample preparation such as drying or vacuum conditions. Additionally, large numbers of individual particles must be characterized to estimate the properties of the entire sample. Few recent reviews related to the application of novel microscopic tools for imaging of different nanomaterials are currently available [141,142,143], but none of them focuses specifically on the silica particles.

Transmission electron microscopy (TEM) and scanning electron microscopy (SEM) are the most popular microscopic tools for the characterization of nanomaterials. TEM, based on an electron beam sent through a very thin slice of the sample, is capable of imaging with much higher resolution than light microscopes, thus allowing observation of much finer details. SEM creates images of the sample by scanning the surface with a focused electron beam and visualizes details on the surface of the samples, providing a 3D view. Both methods are widely utilized for MSNs characterization and can be used separately or might be combined to visualize the internal structure of particles as well as their morphologies. In recent years, more sophisticated electron microscopy techniques have been applied for MNPs imaging. High-resolution transmission electron microscopy (HR-TEM) is an imaging mode of specialized transmission electron microscopes that allows for direct imaging of the atomic structure of samples. It can be used to accurately determine the nanopore size. High resolution methods have been applied to check the pore size and morphology, as well as its stability after modification and guest incorporation [15]. High-angle annular dark-field scanning transmission electron microscopy (HAADF-STEM) is a TEM method that receives inelastically scattered electrons or thermal diffuse scattering (TDS) at high angles using an annular dark-field (ADF) detector. Its resolution is nearly determined by the incident probe diameter on the sample. A HAADF-STEM instrument provides an exceptional resolution, higher than 0.05 nm, and is highly sensitive to variations in the atomic number of atoms in the sample. A combination of HRTEM and HAADF-STEM can be used for in-depth characterization of nanoparticles that have a high pore volume while having a low outer surface area, as was shown by Huand et al. (Figure 9). Moreover, the application of the fast Fourier transform (FFT) of the high-resolution images, combined with the statistical analysis of images, revealed details regarding the pore sizes, their ordered structure, and integrity. The results were confirmed by applying the HAADF-STEM imaging over an extended tilt range acquiring images for every 2° of tilt. The contrast of obtained images depends solely on the projected mass-thickness, making it an ideal tool for visualizing the structure of the internal pore channels, and finally confirmed that individual pores run through the length of the particle and are arranged in an ordered structure [144].

HAADF-STEM was also used in several other recent works for detailed characterization of MSNs, especially to observe the specific morphology and to identify the position of loaded cargo [145,146,147]. For the determination of the structural characteristics and the localization of deposited phase within the mesoporous matrix, the incoherent HAADF-STEM imaging mode was applied on a few specimen fragments. It visualized the high organization of the mesopores, their long-range ordering, and preservation [148]. In the same work, the TEM-EDX and STEM-EELS/EDX have been performed for the chemical analysis of samples to further proved the presence of Mo in the bright areas and of S in the MoS_2_ structures. Electron microscopy (TEM or SEM) combined with the energy-dispersive X-ray spectroscopy (EDX or EDS) or electron energy loss spectroscopy (EELS) is a tool used to identify the elemental and compositional analysis of the samples, regarding both, the compositional information of the whole sample, as well as the location of individual elements. The dark-field TEM imaging and corresponding EDS elemental mapping were applied as well to follow the distributions of various elements and loading of gold nanorods into dual-mesoporous silica spheres (DMSSs) [149]. TEM immunogold labeling (IGS-TEM) of ultrathin sections was developed to study the binding of human lysozyme on SBA-15 mesoporous silica, showing their adsorption on both the external and the internal pore surface. Additionally, a three-dimensional reconstruction of the particle from TEM micrographs was also implemented to improve the visualization of lysozyme distribution [150]. In the following studies, the technique was improved by the silver enhancement procedure, allowing the increase in the size of ultrasmall gold nanoparticles (GNPs), which enables the direct imaging of the absorbed conjugates [151,152]. Visualization and quantification of enzyme immobilized in MSNs have been facilitated by the combination of nitrogen sorption analysis and IGS-TEM, allowing direct imaging of guest distribution inside the particles [153].

An optical microscope uses visible light and a system of lenses to generate magnified images of the sample. The basic microscope is a very simple instrument, but many complex designs have been developed over the years to improve resolution and contrast. In the characterization of nanoparticles, particularly those carrying fluorescent cargos, confocal microscopy is especially applicable. Super-resolution (SR) fluorescence microscopy is a class of optical microscopy techniques at a spatial resolution below the diffraction limit. Stochastic optical reconstruction microscopy (STORM) is a widely used type of SR technique, able to reconstruct a super-resolution image by combining the high-accuracy localization information of individual fluorophores in 3D and multiple colors. It is based on a stochastic switching of a single fluorescent molecule using very low-intensity light, which by repetition through numerous cycles allows the fluorophores mapping with high precision. Although STORM has been used mostly in biological studies, it is also a great tool for experiments at the intersection of biology and materials science, and it was applied to study the penetration of proteins within the porous silica nanoparticles [154]. In comparison to the standard wide-field methods with a diffraction limit of 250 nm, STORM offers impressive improvement of performance (Figure 10).

Recently, STORM was employed for direct observation of nanoparticles within cells with high spatial resolution [155]. In order to characterize processes in situ, confocal laser scanning microscopy (CLSM) combined with microfluidics (MFs) was applied for monitoring of protein adsorption onto silica microparticles. The technique is based on monitoring the increase in fluorescence intensity from the adsorption of fluorescently labeled proteins onto the particles. It offers valuable information about formation kinetics and protein stability. The combination of MFs with high-speed camera imaging allowed detailed insights into very small time scales (~0.5 s) of protein corona formation [156].

Heating optical microscopy is used for the characterization of the drying behavior of droplets containing solids or solutes. The understanding of the dynamic process of drying is usually difficult but important for various nanotechnological applications, including the preparation of nanostructures. The optical microscope with a heating and vacuum stage has been used for the investigation of controlled drying of monodisperse emulsion droplets containing colloidal silica nanoparticles and their subsequent assembly into mesoporous silica microspheres (MSMs) [157].

Atomic force microscopy (AFM) is a type of scanning probe microscopy that uses a fine tip to probe surfaces and has a very high resolution. It is one of the most popular techniques for surface topography determination and nanostructures investigation. AFM can be applied to visualize almost any type of surface, constituting it as one of the most powerful microscopy tools for analyzing samples at the nanoscale. Moreover, AFM does not only provide screening of surface topography but may contribute significantly to the understanding of materials’ properties in terms of mechanical, physical, thermal, and chemical features, by application of advanced tools such as phase image AFM, conductive AFM, or thermal analyses [158]. In general, three different AFM modes are used: contact mode, non-contact mode, and tapping mode. In contact mode, the tip makes direct contact with the surface of the sample. It is robust, but quite invasive and may lead to the displacement of objects on a surface, and even cause damages to the sample and the tip. In non-contact mode, an oscillating probe is moved close to the surface, and the tip never makes direct contact with the sample. This mode is usually used in (ultra) high vacuum and can achieve very high resolutions, down to the atomic level. The tip never touches the sample and therefore cannot disturb or destroy it. Tapping mode (or intermittent contact mode) is similar to non-contact mode and uses an oscillating cantilever as well, but the tip “taps” on the surface during the oscillations. This mode is much less invasive than contact mode and can be used under atmospheric conditions. It was recently shown that the AFM technique can be applied to visualize protein aggregation at the single-molecule level [159]. Based on that, AFM images have been applied as a criterion to determine the aggregation state of xylanase before and after immobilization on organically modified biogenic MSNs, by following the heights of molecules in topographical imaging [160]. AFM imaging in the liquid phase was applied to investigate the MSNs templated self-supported HSA nanocapsules loaded with doxorubicin, revealing their shape, size, aggregation, and homogeneity [161].

Photoacoustic microscopy (PA) is an imaging method based on the photoacoustic effect and uses the rise of local temperature occurring as a result of light absorption in tissue. It has a wide range of applications in the biomedical field and recently has been used to follow the trafficking of nanoscale materials inside cells. The encapsulation of PA imaging agents inside functionalized MSNs has been studied to obtain the most efficient PA imaging performance [162,163]. Thus, there is a potential to use PA methodology in the future to study drug release in vivo relying on its co-delivery with PA imaging molecules.

Most of the microscopic techniques discussed above have also found applications in the study of the interaction of low molecular weight drugs with mesoporous silicon matrices. The literature cited here is proof of this [74,152,164].

### 3.4. Spectroscopy

#### 3.4.1. Solid-State NMR

In the treatment process, most drugs are administered in the form of oral solid tablets. The action of such substances in the body, the biological activity, and bioavailability depends on the physical state of APIs (polymorphic forms, solubility) and accompanying substances (preservatives or auxiliary substances). Certainly, the study of molecular and physicochemical properties should be carried out with the use of tools that provide the broadest possible spectrum of structural information on condensed matter. Solid-state NMR (SS NMR) spectroscopy fulfills these conditions [165].

Taking into account that over 40% of drugs on the market are poorly water-soluble, which significantly limits their bioavailability, improving this parameter is extremely significant [166]. It is no wonder then that the use of mesoporous silica particles as solubilizers is of increasing importance [167,168,169]. From the pharmaceutical standpoint, drug-carrier interactions, amount (degree of loading), location, local molecular dynamics, and physical state of the loaded drug are important. SS NMR is a non-destructive, highly flexible technique which allows the determination of various parameters. By carrying out different experiments structural information and dynamic parameters can be obtained during one measurement session. Contrary to other techniques, SS NMR can be applied to any solid physical state (both crystalline and amorphous) and materials of different complexity, from pure APIs or excipients to solid dispersions (including commercial preparations) [165]. Moreover, the development of NMR methodology and the use of new experiments, especially fast magic angle spinning (MAS) [170,171,172,173], dynamical nuclear polarization (DNP), and surface-enhanced NMR spectroscopy (SENS) [174,175,176], makes the SS NMR the premier method in the study of mesoporous materials, drugs confined in nanocarriers and API-MSN interactions [170,177,178].

Since the pioneering work published by Vallett-Regi et al. [1], a lot of papers on drug encapsulation in mesoporous silicas, mainly MCM-41, SBA-15, and their various modifications appear regularly, where SS NMR is used [136,179,180,181,182,183,184,185,186,187,188,189,190]. In many of them, ibuprofen is often employed as a model molecule for the study of host-guest interactions. Ibuprofen is a common nonsteroidal anti-inflammatory drug (NSAID) that has a well-defined structure and physical properties (melting point, polymorphism, etc.). For example, Babonneau et al., presented studies of ibuprofen encapsulated into MCM-41 and MCM-41 functionalized by amino groups [179]. Using single pulse (SP) and cross-polarization (CP) ^13^C MAS NMR techniques, as well as ^1^H MAS NMR experiments, they showed extremely high mobility of ibuprofen molecules when the silica was not functionalized. In contrast, when the MCM-41-NH_2_ silica matrix was used, the ^13^C NMR experiment indicated more limited mobility of ibuprofen, which confirmed interactions between silica amine groups and carboxyl groups from drug molecules. A more advanced and complete SS NMR study of ibuprofen encapsulated in MCM-41 silica with different pore diameters (35 Å and 116 Å) was presented by Azais et al. [180]. Its behavior was investigated using ^1^H, ^13^C, ^29^Si multinuclear SS NMR experiments at ambient and low temperatures. They demonstrated that ibuprofen was extremely mobile and behaved like a liquid inside the pores at ambient temperature. This unexpected physical state of the encapsulated molecules in mesoporous silica was explained by the existence of the confinement effect [128,183,191,192,193], which was often studied on simple liquids such as water [194,195], methanol [196,197], or benzene [198,199,200]. The ^1^H–^13^C and ^1^H–^29^Si heteronuclear correlation (HETCOR) experiments proved that there was a weak interaction between the drug and the silica, which favors fast drug release from this material in simulated biological fluids. Moreover, the SS NMR study showed that the carboxylic proton of the ibuprofen was in a chemical exchange at room temperature with other protons, possibly from silica walls (SiOH) and/or water [180]. In the latter paper, the same group demonstrated examining ibuprofen and benzoic acid encapsulated in MCM-41 through incipient wetness impregnation method using variable temperature ^1^H MAS NMR experiments [181]. The authors confirmed the presence of water molecules inside the pores of silica and their participation in the fast chemical exchange with protons of the COOH group of organic molecules at room temperature. Additionally, a study by Tielens et al., combining SS NMR with molecular dynamics calculations, proved that the presence of a monolayer of water surface molecules on silica plays the role of a “glue” permitting high mobility of ibuprofen and benzoic acid molecules [201]. In another study conducted for the ibuprofen/MCM-41 system, comparing the effectiveness of incipient wetness impregnation and melting methods by analysis of the filling factors, it has been shown that the affinity of ethanol used in the impregnation process for mesoporous silica significantly decreased this parameter [83]. The percentage of filled mesopores was determined by analysis of ^1^H Very Fast (VF) MAS NMR spectra recorded at a spinning rate of 60 kHz. The obtained results presented on the performance profiles of both procedures showed that the filling factor for the melting method was much higher compared to incipient wetness impregnation and obtained at 60% and 20%, respectively (Figure 11). Moreover, the stability of the ibuprofen/MCM-41 sample prepared by the melting method in the environment of ethanol and water vapor was investigated. Using advanced SS NMR experiments it has been confirmed that ethanol vapor can remove ibuprofen from MCM-41 pores. When the sample was stored in a closed vessel filled with ethanol vapor for several hours, the ibuprofen was transported to the external walls of the silica. The action of water vapor was very similar, but the penetration of water into the pores and the exchange of ibuprofen were slower compared to the analogous procedure using ethanol. This study demonstrated that both solvents acted like “molecular pistons” pushing ibuprofen from the pores [83].

The structural and dynamic properties of indomethacin molecules embedded within the mesopores of SBA-15 and SBA-15 functionalized with 3-aminopropyltriethoxysilane (APTES) using ^1^H, ^13^C, ^29^Si MAS NMR spectroscopy, and ^1^H NMR spin-lattice relaxation measurements were investigated by Ukmar et al. [182]. The conducted studies showed that only when the concentration of indomethacin within the mesopores was high the hydrogen bonds between drug molecules were responsible for dimer formation, while at low concentrations indomethacin formed a layer on the silicate walls. Moreover, 2D NMR correlations revealed that the interaction between the indomethacin molecules and the silicate surface was moderate to weak. When indomethacin was incorporated into the functionalized SBA-15 matrix, drug-matrix interactions were enhanced. Skorupska et al. used advanced SS MAS experiments to analyze other systems of pharmaceutical co-crystal loaded into mesoporous silica [202,203]. The authors used very fast MAS NMR, ^1^H detected ^1^H–^13^C HETCOR and ^1^H–^15^N HETCOR 2D correlations measurements, in combination with 2D ^1^H–^1^H BaBa to characterize the loading process of naproxen/picolinamide and ibuprofen/nicotinamide co-crystals to SBA-15 and/or MCM-41 mesopores and their interactions. Lately, co-crystals have become very popular among scientists and the pharmaceutical industry [204,205,206], and the issue of co-crystal embedded in MSN has not been described in the literature until 2015.

Very recently, the applicability of the new DiSupLo loading method employing selected APIs—ibuprofen, ketoprofen, flurbiprofen, and MCM-41 mesoporous silica, was confirmed by Potrzebowski et al. using advanced SS NMR experiments [47]. The ^1^H NMR spectra recorded with sample spinning of 60 kHz, ^13^C CP-MAS, ^13^C single pulse experiment (SPE) and 2D correlations ^1^H–^1^H RFDR (Frequency Driven Dipolar Recoupling) MAS, ^1^H–^13^C invHETCOR), ^1^H–^13^C HSQC High Resolution (HR) MAS were applied to analyze the loading process and APIs interactions inside the pores. The ^13^C MAS spectra showed that APIs were located inside the MCM-41 pores when the weight ratio of API to MCM-41 was 1:3 and 1:2. The efficiency of CP significantly decreased due to the high mobility of drug molecules entrapped in the silica matrix, therefore SPE was appropriate. When the API/MCM-41 ratio was changed to 1:1, the CP-MAS spectra showed additional signals indicating the position of the API both inside and outside the MCM-41 pores. The ^13^C NMR and ^1^H VF MAS spectra were compatible. The peak of the acidic proton of API was not visible in the ^1^H VF NMR spectra when the whole drug was in the pores. The acid hydrogen of API located in the pores is involved in the interaction with oxygen and MCM-41 hydroxyl groups, forming new C–O–H ... O–Si hydrogen bonds. The transfer of API carboxylic proton on the MCM-41 walls was confirmed by comparing the chemical shifts of ^13^C and ^1^H for the crystalline and loaded ibuprofen taken from the 2D ^1^H–^13^C invHETCOR (Figure 12). The comparison showed that the downfield shift of the methine proton signal was the effect of carboxyl proton transfer and can be considered a measure of the efficiency of the API loading process. Moreover, it has been proven that the DiSupLo method allows quantitatively loading a mixture of two or three APIs into the pores of MCM-41 [47].

Progress in solid-state NMR studies of drugs confined within different drug delivery systems has been presented by Skorupska et al. [207]. Recently, Potrzebowski et al. have presented the use of SS NMR to study pharmaceuticals also API loaded into mesoporous DDS, extensively discussed modern NMR techniques that are used in structural research of pharmaceutical products, including techniques using fast and very fast magical angle spinning with sample rotation over 40 kHz [208]. Li et al. reviewed the advancement and application of SS NMR in pharmaceutical sciences to overcome challenges in material characterizations [209].

#### 3.4.2. Vibrational Spectroscopy

Vibration spectroscopy techniques such as infrared and Raman spectroscopy compared to other methods discussed in previous sections offer specific advantages in a variety of applications. Even though both techniques are completely different from each other, they work in a complementary way. It is worth emphasizing that these two spectroscopies do not probe the same vibrational information of the molecule. Raman spectroscopy is based on an inelastic scattering process and detects vibrations related to a change in polarizability, while infrared spectroscopy is based on an absorption process and detects vibrations related to a change in dipole moment. As a result, the two spectroscopies have different selection rules and it is possible that a strong vibration mode in the Raman spectrum may be weak in the IR spectrum or not be visible, and vice versa. Moreover, in Raman spectroscopy, a compound can be tested in an aqueous solution, while in infrared spectroscopy it is a serious limitation due to the excessive adsorption of water. Infrared and Raman spectrometers are divided into two categories: dispersion and Fourier transform.

*Infrared (IR) spectroscopy* is a non-destructive, easy-to-use universal research tool for examining the chemical status of porous silica surfaces. The convenience of this method is due to the transparency of silicon for infrared light. The basis of the IR analysis of mesoporous silica is the knowledge of the position of the absorption bands of Si–O bonds. The silica matrix is reflected in the IR spectra by (a) bands of stretching vibrations in the range of 3700–3400 cm^−1^ isolated hydroxyl groups (sharp band shape) and/or derived of absorbed water (broadly absorbing band, confirmed by a strain band at 1620 cm^−1^), (b) most strongly absorbing bands of Si–O–Si stretching vibrations observed at 1200–1000 cm^−1^ (asymmetric Si–O–Si stretch), (c) multiple absorption bands below 1000 cm^−1^ characterizing, for example, symmetrical Si–O–Si stretching vibrations (at 790 cm^−1^), Si–O–Si bending vibrations (at 460 cm^−1^) [210,211,212]. In recent years, infrared spectroscopy has become an indispensable technique for pharmaceutical analysis. The infrared spectrum represents absorption and molecular transmission, creating a molecular fingerprint. The material is a unique combination of atoms, and no two compounds produce the same infrared spectrum. Changes in the frequency and shape of the drug bands in the IR spectrum can be used to analyze the electron density redistribution in the structure of the molecule to evaluate interactions [213]. The dynamic development of instruments and the use of the Fourier transform in the analysis of the recorded signal allowed for a significant reduction in the measurement time of a single IR spectrum and the development of FT-IR spectroscopy (infrared spectroscopy with Fourier transform spectroscopy). This technique is currently the most widely used to characterize the surface chemistry of mesoporous silica, both with surface modified [214,215] and drug-loaded. Changes in the appearance of functional groups on the FT-IR spectrum of mesoporous silica and drugs illustrate the structural changes that occur during the loading process and the interactions that occur between the studied systems. If the drug interacts with mesoporous silica, the drug band becomes smaller or less sharp compared to the pure drug spectrum. In other words, a reduction in the intensity of the drug functional group bands loaded into the mesoporous system indicates a strong molecular interaction between the drug and silica. In the case of physical interaction, distinct drug and silica bands appearing in the spectrum indicate that the drug has been adsorbed on the outer surface [7]. Most used in research study model drug—ibuprofen in the FT-IR spectrum shows a characteristic intense band at 1711 cm^−1^ for the stretching vibrations of the carbonyl group (C=O) and three bands in the range of 3300–2800 cm^−1^ corresponding to the stretching vibrations of the benzene ring (C–H), Figure 13 [47,216].

Li et al. investigated the effect of ibuprofen at various concentrations on the surface of two disordered mesoporous materials: Syloid 244FP and Neusilin NS2. In the FT-IR S244FP and ibuprofen spectra, all tested drug to silica ratios show bands characteristic of pure substances, which proves the physical adsorption of ibuprofen on the outer surface of mesoporous silica. In turn, in the samples of ibuprofen loaded into NS2, the carbonyl group was reduced to 1591 cm^−1^, which indicates strong intermolecular interactions of IBU: NS2 [217]. In the discussed case, the carbonyl group reacted with the OH group of MgO and Al_2_O_3_ groups located on the surface of the pores of NS2, forming salts. On the other hand, SiO_2_ groups of silica did not participate in the reaction, so the bands characteristic for this group at 1300–900 cm^−1^ and 3700–3000 cm^−1^ did not change in the FTIR spectrum. Tzankov and co-workers presented the encapsulation of pramipexol in MCM-41 mesoporous silica particles. The FTIR spectrum of empty MCM-41 showed a band at 1047 cm^−1^ corresponding to the Si–O–Si stretch vibration. The spectrum of the pure drug showed absorption bands at 3410 (N–H stretching vibration), 2945 (C–H stretching vibration), 1586 (C=C stretching vibration), 1309 (C–N stretching vibration), and 760 cm^−1^ (C–H bending vibration). On the other hand, the FTIR spectrum of particles loaded with pramipexol revealed bands at 1520 cm^−1^ characteristic of secondary amines and bands of 1585 and 1633 cm^−1^ for primary amines. The spectrum also shows the 2972 cm^−1^ band, indicating the presence of the NH_3_^+^ group and indicating interaction with silanol groups of the silica [218].

*Raman spectroscopy* of porous silica can provide valuable information on the properties of silicon nanostructures, which are strongly dependent on symmetry, structural geometry, morphology, pore diameter, backbone size, etc. The first-order phonon entrapment model of optical phonons at 521 cm^−1^ (calibration of spectrometer) is often used for shape analysis Raman scattering of the porous silicon bands, thereby determining the size of the crystallites embedded in the porous layer. The nano-sized particles also exhibit low-frequency acoustic vibrations, which can be observed by Raman spectroscopy. Pure MCM-41 on the Raman spectrum has characteristic bands at about 918, 542, and 126 cm^−1^ assigned to Si–OH, six or four-membered ring vibration, and bending mode of Si–O–Si vibrations, respectively [219]. Using UV resonance Raman spectroscopy, it was estimated that sample MCM-41 showed four weak bands at 499, 595, 798, and 982 cm^−1^. The 499 and 595 cm^−1^ bands are assigned to the asymmetric and symmetrical ascending vibrations of the Si–O–Si bond, respectively, and the 798 cm^−1^ bands are assigned to the vibrational mode of the siloxane bond. The 982 cm^−1^ band is related to the Si–O–Si bond which is directly related to the skeleton defects. In the case of surface modification of MCM-41 with sulfated zirconium, an increase in the intensity of this band was found [220]. Similar relationships were shown in the study of the mesoporous SBA-15 surface loaded with vanadium [221]. The much higher sensitivity of Raman spectroscopy compared to IR for groups such as the thiol group made this spectroscopy the basis for determining the cisplatin loading of mesoporous silica modified with SH groups [222]. Bouyer et al. observed vibration bands of tetracyclosiloxane rings—486 cm^−1^, symmetric stretching bands from the Si–O–Si group—800 cm^−1^, and vibrations related to the stretching mode of Si–OH hydroxyl surfaces—978 cm^−1^ for both MSN–SH and MSN. In addition, the band assigned to the S–H binding at 2581 cm^−1^ was identified in the MSN–SH spectrum. In turn, pure cisplatin on the Raman spectrum shows four characteristic bands attributed to the deformation of the skeleton in the bond plane: a. Cl–Pt–Cl (162 cm^−1^), b. N–Pt–N (255 cm^−1^), c. Pt–Cl (322 cm^−1^) and a band in the Pt–N stretching mode (524 cm^−1^). In the mesoporous silica-drug system, pure cisplatin bands are not revealed, which confirms the presence of the drug in the pores of the silica. On the other hand, a wide band of vibrations was observed at 310 cm^−1^, which corresponds to the Pt–S stretching mode and confirms the coordination of cisplatin by thiol groups, Figure 14.

The introduction of the poorly soluble drug indomethacin into mesoporous silica MCM-41 and SBA-15 first using imaging by Raman spectroscopy was discussed by Hellstem et al. [223]. Raman imaging is particularly useful in the pharmaceutical industry because determines the spatial distribution of compounds in a sample and is sensitive to the crystallinity and polymorphism of compounds. The study of polymorphism is especially important because it relates directly to the stability and safety of the drugs [224]. The Raman spectra of solid indomethacin forms present in the silica particles are shown in Figure 15. The characteristic band for α-indomethacin loaded silica occurs at the wavenumber of 1650 cm^−1^ and for γ-polymorph at 1699 cm^−1^. The spectrum of amorphous indomethacin has a broad band around 1681 cm^−1^. As shown in Figure 15, the bands of these three forms partially overlap, which makes the spectra interpretation more demanding. The authors used a partial least-squares analysis of the Raman spectra, thus determining the solid form of the loaded drug. Figure 15 presents the uniform distribution of the two indomethacin alpha and gamma polymorphs in the carriers, while the increase in temperature and humidity leads to the formation of clusters with higher crystallinity.

Despite the problems encountered during the analysis: fluorescence and combustion of the samples, this study provided useful information on drug distribution and the presence of unexpected solids such as solvates or rare polymorphs of the model drug.

The combination of Raman scattering anti-Stokes microscopy (CARS) with CARS hyperspectral analysis is a powerful tool capable of selectively imaging drugs incorporated into mesoporous silica-based materials. The CARS variant microscopy was used for the first time by Offerhaus et al. [223,224,225] to observe differences in the distribution of drugs (itraconazole and griseofulvin) and the physicochemical structure of ordered mesoporous silica particles.

### 3.5. Powder X-ray Diffraction (PXRD) and Small-Angle X-ray Scattering (SAXS)

In order to understand the structural characterization of nanocarriers different X-ray scattering techniques might be applied revealing information about their crystal structure, chemical composition, and physical properties. It is especially important for the development of drug delivery systems, where there is a strong structure-function correlation.

In powder X-ray diffraction (PXRD) the pattern is obtained from a bulk matter, rather than an individual crystal as happens in the case of single-crystal X-ray diffraction. Not only each API will produce a specific pattern depending on the structure of its crystal lattice but even each polymorph, salt, or co-crystalline material will have its characteristic diffractogram. PXRD can be used to determine if any change in drug form has occurred during the loading process [226]. The absence of crystalline peaks with sufficient quantity indicates that the material is amorphous. For example, it was shown by a combination of DSC and PXRD that mesopores change the insoluble crystalline form of angiotensin II receptor blocker called valsartan to a soluble amorphous state [227]. Similarly, the PXRD patterns of untreated drugs: ibuprofen, ketoprofen, and flurbiprofen have shown their crystalline structure, while the absence of APIs peaks in the drug-loaded MCM-41 diffractograms indicates that the drugs are loaded inside the pores of the carriers and that they are amorphous (Figure 16) [47].

PXRD is widely used for the MSNs characterization and there are many recent literature examples for its application. For example, to determine the crystalline structure of the magnetic nanoparticles before and after silica coating [228], or to confirm the presence of iron oxide and identify the relative crystalline phases formed in the synthesis of polymer grafted MSNs [229]. Further cases show the loading of indomethacin in an amorphous phase [230], and characterization of the mercapto-functionalized MCM-41 proving their stability after the modification process [222]. Importantly, one has to realize that the data provided by XRD experiments arises from the structural properties of crystal domains and not necessarily from the entire particle. Thus, when analyzing particle sizes, the use of different methods, such as microscopy, is necessary to obtain reliable information.

Small-angle X-ray scattering (SAXS) provides information about the structural characteristics of nanomaterials. It determines nanoparticle size distributions, resolves the size and shape of macromolecules, verifies pore sizes, etc. For successful application of SAXS, the prior study of the sample preparation method, particle morphology, size distribution, aging stability, etc., is required via other approaches, such as electron microscopy, dynamic light scattering, and zeta potential. SAXS has been applied, for example, in the characterization of the amino-functionalized mesoporous silica nanoparticles that display the typical pattern of hexagonal mesoporous structures, with an intense peak, due to the reflection of 1 0 plane, and two weak peaks due to the reflection of 1 1 and 2 0 planes, respectively. The value of lattice parameter, a, has been obtained as well [231]. SAXS analysis supported characterization of modified and unmodified silica nanoparticles, before and after loading with naproxen. Three resolved reflections were attributed to the well-defined mesoporous structure with two-dimensional hexagonal p6mm symmetry, while the samples with larger heterogeneity of the surface exhibited lower scattering power [15]. It was applied also in some other recent drug loading studies [184,232], and to characterize in detail the inner structures of MSNs, providing information about their homogeneity, average mass density of spheres, and the volume fraction of silica [233]. It is worth mentioning that SAXS is also a widely used method for MSNs preparation. For the first time, the growth and kinetics of MSNs were investigated by time-resolved SAXS with synchrotron irradiation [234]. Nucleation and growth of SBA-15 silica particles have also been studied by SAXS and USAXS (ultrasmall-angle X-ray scattering). In comparison to SAXS that measures sizes related to the micelles and mesoscopic ordering, USAXS probes length scales up to a few micrometers. Thus, the combination of both methods gives information on all length scales associated with the system [235]. Moreover, wide-angle X-ray scattering (WAXS) might be occasionally applied to characterize the amorphous structure of synthesized MSNs [236].

### 3.6. Chromatography

High performance liquid chromatography (HPLC) is used to separate, identify, and quantify different mixture components, and it is applied to confirm the chemical stability of the molecules loaded into nanoparticles [237,238], entrapment efficiency [239,240], and to obtain the drug release curves, as demonstrated in the release studies of naproxen [15], paclitaxel [87], resveratrol [241], felodipine and furosemide [17], and many others. For the determination of drug loading efficiency, after the completion of the loading process, samples undergo centrifugation and the unloaded molecules present in the supernatant are analyzed by HPLC, as was shown in the case of rifampin [242].

### 3.7. Mass Spectrometry

Secondary ion mass spectrometry with a time-of-flight analyzer (ToF-SIMS) is a destructive and extremely sensitive technique for analyzing the surface composition and depth profiling of organic and inorganic materials. The test method consists of etching the surface of the sample by bombarding with a beam of primary ions (Cs or micro-focused Ga), and then by carrying out a mass analysis of the ionized matter, information on the elemental composition of the tested sample is obtained [243]. Along with the etching of successive layers of the material, information is obtained regarding the change in the composition of the sample as a function of depth (the so-called deep profile). TOF-SIMS analysis provides information in the sample about 1–1.5 area, with high mass resolution up to 10,000 m/Δm and spatial resolution for 1-micron imaging. Thanks to it, it is possible to determine the levels of admixtures, assess the stability of layers, determine diffusion parameters and describe other phenomena occurring during the production and processing of samples [244]. Kempson et al. [245] used TOF-SIMS to map the distribution of methylene blue and papain in silicon layers. Methylene blue was used as a small molecule drug simulant. The mass of the dye—284 g/mol is within the range of primary masses of the Ga TOF SIMS ion beam and can be easily and unambiguously identified. Papain represented the larger molecules of pharmaceutical importance. It contains 212 amino acid units and forms two domains. Due to its high molecular weight of 23,406 g/mol, only fragmentation products can be detected in TOF-SIMS. Smaller organic papain fragments were compared to the methylene blue decomposition to investigate similarities and differences in loading. Secondary ion mass spectrometry revealed evidence of Si–O, Si–C bonding, it also showed decomposition of organic matter after deposition and loading behavior, literally observing where the organic molecules were. Porous structures are a good medium for loading small and large organic molecules. Load dynamics can be studied as a function of time, concentration, and pore structure. Additionally, detecting organic, inorganic, and molecular fragments from TOF-SIMS, together with complementary data obtained from other spectroscopy, can help better understand the chemical mechanisms of adsorption (Figure 17).

### 3.8. Other Methods

*Dynamic Light Scattering (DLS)* provides information about the size distribution of particles and populations of particles and gives insights into the system’s agglomeration. The main weakness of the method lies in its low resolution and inability to handle highly concentrated samples. Moreover, DLS works only on optically clear particle suspension based on the known or well characterized kinematic viscosity of the solvent. In few MSNs studies, hydrodynamic diameters of the nanoparticles were determined by DLS to show both frequency distribution and cumulative particle-size distribution [228] and mean hydrodynamic diameter [246]. DLS together with static light scattering (SLS) has been applied to characterize size, stability, and porosity of porous silicon and silica nanoparticles [247].

*Zeta-potential (ZP)* measures the electrostatic or charge repulsion or attraction between nanoparticles and can be applied to study the colloid stability and surface charge. The electrophoretic mobility of the particles in drug delivery studies is mostly measured by electrophoretic light scattering. The literature often states NP-dispersions with ZP values of ±0–10 mV, ±10–20 mV and ±20–30 mV and >±30 mV as highly unstable, relatively stable, moderately stable and highly stable, respectively [248]. However, the results are not straightforward, and it is possible to have stable colloids with low ZP and vice versa. Moreover, it is important to report all the information about the measurement conditions including the medium ionic strength, composition, and pH, since they considerably influence the zeta potential. The surface charge measurements are determined by identifying which electrode the particles are moving towards during electrophoresis [249]. The method is widely used in the preparation of MSNs to ensure stability, the success of modification, and lack of aggregation [228,246,250].

*Contact angle* measurement gives the angle between a solid surface and a droplet of liquid on the surface, providing information about the wettability of a solid by a liquid. In the case of complete wetting, the contact angle is 0°. The solid is wettable in the range between 0° and 90°, and not wettable above 90°. In the case of ultra-hydrophobic materials, the contact angle approaches the theoretical limit of 180°. Hence, it provides information about the wettability of silica. Contact angles of the silica particle samples were determined from capillary penetration [251]. The method was used for example in the research about the impact of surface hydrophilicity and pore size on water uptake by mesoporous silica [252], to study superhydrophobic MSNs for doxorubicin delivery [253], and to measure the wettability of ciprofloxacin-loaded MCM-based solid films [254].

*Elemental analysis* (CHNS analysis) provides information about the elemental and sometimes isotopic composition of the sample and is applicable to study the chemical modification of surfaces, thus allowing insights into verification of the functionalization process [255].

*Gas pycnometry* is used for measuring the density of the sample, obtained by calculation of the ratio of mass to volume. The method has been applied to determine the amorphous densities of the model drugs and used to calculate the loading capacities of mesoporous silica [129].

*Isothermal titration calorimetry* (ITC) is a technique commonly used to determine the binding affinity and thermodynamics of biomolecules interactions. ITC measures directly the heat released or absorbed during a binding, simultaneously providing information about a range of binding parameters such as entropy, enthalpy, the equilibrium constants, etc. It is relevant in some studies, involving nucleic acid incorporation into MSNs, to determine the nature of DNA or RNA interaction with the silica [256,257].

*Vibration Sample Magnetometry* (VSM) is applied to test the magnetic properties of samples. It is occasionally used in MSNs studies, in the cases of nanoparticles with magnetic properties. It has been applied to obtain the magnetization curves of mesoporous silica hybrid nanoparticles with superparamagnetic features [228].

## 4. Conclusions and Future Perspectives

The application of nanotechnology-enabled drug delivery systems has gained increasing interest over the past few decades and is emerging as one of the key areas for future developments in pharmaceutics. MSNs are one of the most well-studied inorganic nanoparticles for biomedical applications, due to their high drug-loading capacity, tunable size, and high pore volume. They have a great potential to enhance important drug properties, such as solubility, stability, bio-availability, and controlled release. Moreover, they can be easily functionalized, opening an even wider range of potential applications.

In comparison to the MSNs, non-ordered porous silicas, do not have a uniform and narrow pore size distribution, but disordered pore structures. Different types of them have been explored previously, due to the simpler and inexpensive preparation methods [258]. Lately, the interest in the non-ordered silica materials has re-emerged and there are some recent examples of their application in the DDS, for example, different types of Syloid^®^ particles were used as API carriers [7,62,259,260]. Studies focusing on the other silica materials, such as silica xerogels [261,262], silica aerogels [263,264,265,266], silica nanocomposites [267,268,269], etc. have also been published. Non-ordered silicas seem to be an attractive alternative as drug carrier candidates, nonetheless due to the lack of precise control of their critical parameters, they are less favorable for pharmaceutical applications.

Many methods to load drugs onto mesoporous silica have been developed over the years. The choice of the loading process is one of the most crucial steps in the design of the drug delivery platform since it influences the amount of incorporated drugs, their physicochemical properties, distribution, and availability. Even though there are several excellent techniques available, they all display some disadvantages. Solvent-free methods are relatively fast and enable a quite high drug loading, but on the other hand, they exhibit limited penetration of cargo inside the pores, negatively impacting the amount of drug present. In comparison, the organic solvent methods provide much better drug penetration but are very time-consuming and contribute to the loss of a significant part of the drug due to the filtration step [18]. The solubility of API in a pharmaceutically appropriate solvent also limits the efficiency of ‘wet’ methods. Worthy of mention are novel procedures combining the benefits of both groups, such as DiSupLo, based on the API embodiment into mesopores by organic solvent vapors. It is easy, efficient, environmentally friendly, and does not require any special equipment and/or demanding experimental conditions [47]. Another interesting example is co-spray drying, based on the two-step process: the creation of the initial suspension, followed by the diffusion of the drug into the pores. The method is relatively fast and allows easy separation of the loaded product from the solvent [184]. The sol-gel approach allows a simultaneous synthesis of MSNs and their loading with an amphiphilic biologically active compound that works as a template. It was developed for miramistin but might be easily extended by using micelles or vesicles of other active compounds, providing a very effective one-stage production of loaded particles [270]. Unfortunately, even though the above mentioned loading methods are sufficient for laboratory purposes, there is a need to focus on the development of the scale-up processes for simplified and cost-effective commercial manufacturing. It is also important to remember that for future clinical success the loading method needs to provide a homogeneous distribution of incorporated APIs with very high reproducibility.

Advancement in the production of novel specialized MSNs, particularly for drug delivery purposes, demands progress in analytical techniques suited for their characterization. Such methods require extreme sensitivity, precision, and often resolution on the atomic level. There is no one technique sufficient for the complete characterization of particles-based systems and a combination of methods needs to be applied to obtain satisfactory information. Thus, the development of new analytical instruments combining different instrumental techniques into one piece of equipment may provide reliable results in a shorter time resulting in faster and more efficient work. Accurate and non-destructive methods for the quantification of embodied drugs are still in need of further improvement.

The translation of MSNs laboratory results to human clinical trials is still limited, mostly due to the issues with a precise characterization of nanosystems in vivo. Therefore, one of the most important future trends will be novel developments toward nanomaterial characterization in physiological conditions, in order to follow the fate of both carriers and drugs upon their administration into the body. There are many established analytical tools sufficient for the stage of discovery and development, but for the following phases of preclinical and clinical research, they are often uninformative, too invasive, and require laborious and complex data analysis. The product development needs to address those limitations and find methodologies to predict the clinical outcomes of nanoparticle therapy with reliability and accuracy. Some of the other remaining challenges in this area include the design of new cell and animal models of human diseases, labeling and detection methods, mathematical modeling, and computational simulations.

Despite various obstacles, the progress in the field in the last decade demonstrates growing hope for new silica-based drug delivery methods with considerable market potential. Some of the most promising examples come from the currently ongoing or just concluded clinical trials. Mesoporous silica was shown to be safe in a dose up to 9 g/day in male humans according to completed Phase I and Phase II trials [271]. Cornell dots, ultrasmall inorganic optical-PET imaging nanoparticles [272,273,274] were approved for three different human clinical trials for diagnostic purposes that are currently active or recruiting. Above all, we believe that the outstanding potential of MSNs can soon be explored further, taking into account the possibility to load or functionalize them with combinations of molecules to design multifunctional stimuli-responsive drug delivery platforms enabling simultaneous diagnosis and targeted treatment.

## Data Availability

No new data were created or analyzed in this study. Data sharing is not applicable to this article.

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
