# Peer review of "Mesoporous Silica Particles as Drug Delivery Systems—The State of the Art in Loading Methods and the Recent Progress in Analytical Techniques for Monitoring These Processes"

_pharmaceutics, 2021, doi:10.3390/pharmaceutics13070950_

Round 1

Reviewer 1 Report

The authors performed survey of the literature with respect to the use of mesoporous ordered silicas as drug carriers in the drug delivery systems. It is mainly focused on mesoporous ordered silicas, methods of preparation of drug/carrier composites (DCC), and analytical techniques for silicas and DCC characterization. The survey is of interest from a practical point of view. However, the MS needs additional and not weak efforts of the authors to improve the MS toward a level corresponding to high journal standards.

  • Title:Mesoporous Silica Particles as a (?) Drug Delivery Systems— The State of Art in Loading Methods and Recent Progress in Analytical Techniques for Validation of These Processes”. It is unclear what are “these processes”. The title is too long and confused. It could be simpler, e.g., “Mesoporous silicas as effective carriers in drug delivery systems”.
  • Such very important aspects for drug delivery systems (drug carrier performance) as quantitative dependences of drug release (equilibrium and kinetic) vs. pore and particles sizes and molecular sizes, polarity, charge state of drugs (depending on the medium characteristics) should be better described in the survey.
  • What about silica gels, precipitated silicas, fumed silicas, fused silicas, silica aerogels, etc. as drug carriers? It should be compared (at least, briefly) with mesoporous ordered silicas.
  • Figure 7. Derivatives of the weight loss should be negative.
  • Figure 8. There is no sense to show p/p0 (X-axis) up to 1.2 since it is maximal at 1.0. PSD calculated using the BJH method are rather incorrect. Additionally, in the values comma is used instead of dot.
  • Some indexes are not used as subscripts or superscripts.
  • There are errors, misprints and other inaccuracies. The MS needs very careful checking with respect to all aspects.

Author Response

Title:Mesoporous Silica Particles as a (?) Drug Delivery Systems— The State of Art in Loading Methods and Recent Progress in Analytical Techniques for Validation of These Processes”. It is unclear what are “these processes”. The title is too long and confused. It could be simpler, e.g., “Mesoporous silicas as effective carriers in drug delivery systems”.

Answer: The suggestion for changing the title of this review appeared in comments of both reviewers. The reviewer 1 suggests shortening the title to “Mesoporous silicas as effective carriers in drug delivery systems". In opinion of reviewer 2 title should be change on "Mesoporous Silica Particles as  Drug Delivery Systems— The State of Art in Loading Methods and Recent Progress in Analytical Techniques to Characterize"

In fact we have problem how to fulfill the requests of referees. First, when we were invited to submit this review we agreed with Editor that we will describe only part of big problem related with mesoporous silica carriers. There is a plethora review articles in this field. The search of Web of Knowledge (10.06.2021) showed that there is ca 1600 papers with key words"mesoporous silica drug delivery review". In 2020 almost 300 articles were published. Moreover, our search revealed that now authors of reviews rather try to describe part of the problem which is dedicated to specific issue. It was also our intention to locate this article in unexplored area  when we decided to prepare this manuscript.

We agree with referee 2 that word "validation" should be changed. In revised version we propose title;

"Mesoporous Silica Particles as Drug Delivery Systems— The State of Art in Loading Methods and Recent Progress in Analytical Techniques for Monitoring These Processes"

  • Such very important aspects for drug delivery systems (drug carrier performance) as quantitative dependences of drug release (equilibrium and kinetic) vs. pore and particles sizes and molecular sizes, polarity, charge state of drugs (depending on the medium characteristics) should be better described in the survey.

Answer: In our opinion, this comment is very closely related to the suggestion to shorten the title. Under the title "Mesoporous Silicas as Effective Carriers in Drug Delivery Systems," the scope of the review article can be expanded and all aspects shown in Scheme 1,  as drug release (block 4, Scheme 1), as well as drug carrier synthesis (modifications)  and many others could be described. In our manuscript we clearly stated that our intention is describing problems related with loading techniques (block 2, Scheme 1). Other problems (including release) are reviewed elsewhere by other authors  and appropriate references are cited below. Some of them are cited in our manuscript.

  1. Ding, C.; Li, Z. A review of drug release mechanisms from nanocarrier systems. Sci. Eng. C 2017, 76, 1440-1453, doi: 10.1016/j.msec.2017.03.130.
  2. Castillo, R.R.; Lozano, D.; González, B.; Manzano, M.; Izquierdo-Barba, I.; Vallet-Regí, M. Advances in mesoporous silica nanoparticles for targeted stimuli-responsive drug delivery: an update. Expert Opin. Drug Deliv. 2019, 16, 415-439, doi: 1080/17425247.2019.1598375.
  3. Iturrioz-Rodríguez, N.; Correa-Duarte, M.A.; Fanarraga, M.L. Controlled drug delivery systems for cancer based on mesoporous silica nanoparticles.  J. Nanomedicine 2019, 14, 3389-3401, doi:10.2147/IJN.S198848
  4. Doadrio, A.L.; Salinas, A.J.; Sánchez-Montero, J.M.; Vallet-Regí, M. Drug release from ordered mesoporous silicas. Pharm. Des. 2015, 21, 6189-6213, doi: 10.2174/1381612822666151106121419.
  5. Wong, P.T.; Choi, S.K. Mechanisms of Drug Release in Nanotherapeutic Delivery Systems. Rev. 2015, 115, 3388-3432, doi:10.1021/cr5004634.
  6. Andersson, J.; Rosenholm, J.; Lind, M. Mesoporous Silica: An Alternative Diffusion Controlled Drug Delivery System. In Multifunctional Biomaterials and Devices, Ashammakhi, N., Ed.; 2008;
  7. McCarthy, C.A.; Ahern, R.J.; Dontireddy, R.; Ryan, K.B.; Crean, A.M. Mesoporous silica formulation strategies for drug dissolution enhancement: a review. Expert Opin. Drug Deliv. 2016, 13, 93-108, doi:10.1517/17425247.2016.1100165.
  8. Vallet-Regí, M.; Colilla, M.; Izquierdo-Barba, I.; Manzano, M. Mesoporous Silica Nanoparticles for Drug Delivery: Current Insights. Molecules 2017, 23, 47, doi:10.3390/molecules23010047
  9. Manzano, M.; Vallet-Regí, M. Mesoporous Silica Nanoparticles for Drug Delivery. Funct. Mater. 2020, 30, 1902634, doi: 10.1002/adfm.201902634.
  10. He, Q.; Shi, J. Mesoporous silica nanoparticle based nano drug delivery systems: synthesis, controlled drug release and delivery, pharmacokinetics and biocompatibility. Mater. Chem., 2011, 21, 5845–5855, doi:10.1039/C0JM03851B.
  11. Riikonen, J.; Xu, W.; Lehto, V.P. Mesoporous systems for poorly soluble drugs - recent trends. J. Pharm. 2018, 536, 178-186, doi: 10.1016/j.ijpharm.2017.11.054.
  12. Giret, S.; Wong Chi Man, M.; Carcel, C. Mesoporous-Silica-Functionalized Nanoparticles for Drug Delivery. Eur. J. 2015, 21, 13850-13865, doi:10.1002/chem.201500578.
  13. Ramírez-Rave, S.; Bernad-Bernad, M.J.; Gracia-Mora, J.; Yatsimirsky, A.K. Recent Advances in Application of Azobenzenes Grafted on Mesoporous Silica Nanoparticles in Controlled Drug Delivery Systems Using Light as External Stimulus. Mini Rev. Med. Chem. 2020, 20, 1001-1016, doi: 10.2174/1389557519666190904145355.
  14. Abdo, G.G.; Zagho, M.M.; Khalil, A. Recent advances in stimuli-responsive drug release and targeting concepts using mesoporous silica nanoparticles. Emergent Mater.2020, 3, 407–425, doi: 10.1007/s42247-020-00109-x.
  15. Murugan, B.; Sagadevan, S.; Lett J, A.; Fatimah, I.; Fatema, K.N.; Oh, W.-C.; Mohammad, F.; Johan, M.R. Role of mesoporous silica nanoparticles for the drug delivery applications. Res. Express, 2020, 7, 102002, doi: 10.1088/2053-1591/abbf7e.
  16. Kwon, S.; Singh, R.K.; Perez, R.A.; Abou Neel, E.A.; Kim, H.W.; Chrzanowski, W. Silica-based mesoporous nanoparticles for controlled drug delivery. Tissue Eng. 2013, 4, 2041731413503357, doi: 10.1177/2041731413503357.
  17. Karimi, M.; Mirshekari, H.; Aliakbari, M.; Sahandi-Zangabad, P.; Hamblin, M.R. Smart mesoporous silica nanoparticles for controlled-release drug delivery. Nanotechnol Rev. 2016, 5, 195–207, doi: 10.1515/ntrev-2015-0057.
  18. Trewyn, B.G.; Slowing, I.I.; Giri, S.; Chen, H.-T.; Lin, V.S.-Y. Synthesis and Functionalization of a Mesoporous Silica Nanoparticle Based on the Sol–Gel Process and Applications in Controlled Release. Chem. Res.2007, 40, 846–853, doi: 10.1021/ar600032u.

In light of the arguments presented above, we believe that the reviewer will accept our point of view and the decision to select materials for review.

  • What about silica gels, precipitated silicas, fumed silicas, fused silicas, silica aerogels, etc. as drug carriers? It should be compared (at least, briefly) with mesoporous ordered silicas.

Answer: Thank you for this comment. The appropriate discussion is added in conclusions.

  • Figure 7. Derivatives of the weight loss should be negative.

Answer. This Figure is corrected in revised version.

  • Figure 8. There is no sense to show p/p0 (X-axis) up to 1.2 since it is maximal at 1.0. PSD calculated using the BJH method are rather incorrect. Additionally, in the values comma is used instead of dot.

Answer. This Figure is corrected in revised version.

  • Some indexes are not used as subscripts or superscripts.

Answer. It is corrected in revised version.

  • There are errors, misprints and other inaccuracies. The MS needs very careful checking with respect to all aspects.

Answer. It is corrected in revised version.

Reviewer 2 Report

In this review, the authors present an extensive and well written revision of mesoporous silica particles, focusing on the different methods to prepare and to characterize them. In my opinion, the manuscript is suitable for publication, although some aspects may be improved:

  • In the title, I suggest to delete “for validation of these processes” and replace by “to characterize”. In fact, the manuscript describes the different techniques to characterize the nanoparticles, but not how to apply them to validate the system.
  • I have miss a section or examples describing the ability of the mesoporous silica particles to fulfill the objective as drug delivery system and how the different methods of preparation can condition the final result; that is controlled release, increased bioavailability, etc.

Minor comments:

  • Define API
  • Y scale of figure 8c: replace , by .

Author Response

In this review, the authors present an extensive and well written revision of mesoporous silica particles, focusing on the different methods to prepare and to characterize them. In my opinion, the manuscript is suitable for publication, although some aspects may be improved:

  • In the title, I suggest to delete “for validation of these processes” and replace by “to characterize”. In fact, the manuscript describes the different techniques to characterize the nanoparticles, but not how to apply them to validate the system.

Answer: In revised version the title of article is changed.

  • I have miss a section or examples describing the ability of the mesoporous silica particles to fulfill the objective as drug delivery system and how the different methods of preparation can condition the final result; that is controlled release, increased bioavailability, etc.

Answer: This problem is exhaustively discussed in comment to report of reviewer 1. As in previous case,  in light of the arguments presented in comment 1, we believe that the reviewer 2 will accept our point of view and the decision to select materials for manuscript.

Minor comments:

  • Define API

Answer: It is defined in revised version.

  • Y scale of figure 8c: replace , by .

Answer: It is corrected in revised version

Round 2

Reviewer 1 Report

The revised MS could be recommended for publication.